# RSK Isoforms in Acute Myeloid Leukemia

**DOI:** 10.3390/biomedicines9070726

**Published:** 2021-06-24

**Authors:** Minyoung Youn, Jesus Omar Gomez, Kailen Mark, Kathleen M. Sakamoto

**Affiliations:** Department of Pediatrics, Stanford University School of Medicine, Stanford, CA 94305, USA; minyoung@stanford.edu (M.Y.); omarg216@stanford.edu (J.O.G.); kamark@stanford.edu (K.M.)

**Keywords:** RSK isoforms, cancer, hematological malignancy, AML, RSK inhibitors

## Abstract

Ribosomal S6 Kinases (RSKs) are a group of serine/threonine kinases that function downstream of the Ras/Raf/MEK/ERK signaling pathway. Four RSK isoforms are directly activated by ERK1/2 in response to extracellular stimuli including growth factors, hormones, and chemokines. RSKs phosphorylate many cytosolic and nuclear targets resulting in the regulation of diverse cellular processes such as cell proliferation, survival, and motility. In hematological malignancies such as acute myeloid leukemia (AML), RSK isoforms are highly expressed and aberrantly activated resulting in poor outcomes and resistance to chemotherapy. Therefore, understanding RSK function in leukemia could lead to promising therapeutic strategies. This review summarizes the current information on human RSK isoforms and discusses their potential roles in the pathogenesis of AML and mechanism of pharmacological inhibitors.

## 1. Introduction

The Ras-mitogen-activated protein kinase (MAPK) pathway is involved in the regulation of normal cell proliferation, survival, growth, and differentiation [1,2]. More than 30% of all human cancers are associated with abnormal control of this signaling network, resulting in gain of function and subsequent extracellular signal-regulated kinase (ERK) hyperactivation [3]. The Ras-MAPK pathway is initiated by a ligand binding to the receptor tyrosine kinase (RTK) receptor, followed by docking adaptor proteins such as growth factor receptor-bound protein 2 (GRB2) and Son of Sevenless (SOS), leading to activation of the associated Rat sarcoma (Ras) and recruitment of Raf. Raf then activates downstream mitogen-activated protein kinases kinase (MEK1/2) and ERK1/2 (Figure 1) [2].

RSKs are a group of serine/threonine kinases that function in the MAPK signaling cascade and are the direct downstream effectors of ERK1/2. Four RSK isoforms are directly activated by ERK1/2 in response to extracellular stimuli including growth factors, hormones, and chemokines [4,5]. RSKs phosphorylate many cytosolic and nuclear targets resulting in the regulation of diverse cellular processes such as cell proliferation, survival, and motility. Therefore, the RSK isoforms represent attractive therapeutic targets for cancer [6,7]. Here we review the general roles of RSK isoforms and discuss their potential roles in AML and current pharmacological tools to inhibit their function.

## 2. RSK Isoforms

Four RSK isoforms (RSK1-4) have been identified in humans (Figure 2). They share 75–80% homology in amino acid sequence and several conserved functional motifs. Two distinct functional kinase domains are connected by conserved linker regions (turn motif and hydrophobic motif) and are flanked by N- and C-terminal tails [8,9]. The C-terminal kinase domain (CTKD) is involved in auto-phosphorylation of N-terminal kinase domain (NTKD) [9,10,11], and the NTKD is responsible for phosphorylation of downstream substrates. The C-terminal tail contains two conserved motifs: the kinase interaction motif (KIM) for ERK1/2 docking site [12,13,14] and the type 1 PDZ domain-binding motif for the interaction with PDZ domain-containing proteins [15]. RSK3 possesses a potential nuclear localization signal in N-terminal tail.

Inactive RSK isoforms are localized in cytoplasm. Upon stimulation, RSKs are phosphorylated by ERK1/2 and translocate into nucleus for the induction of immediate-early gene expression. RSK4 is distinct from RSK1-3 as it is constitutively active, predominantly localized in cytosol, and exhibits growth-factor-independent kinase activity [16].

RSK isoforms are ubiquitously expressed in every human tissue and brain region with the exception of RSK4, which is mostly expressed during embryonic development [4,17]. This supports the current view that RSKs are functionally redundant. However, tissue-specific variations in expression levels of RSK isoforms have also been reported, suggesting that they might have isoform specificity in mediating distinct cellular functions [4,18,19]. RSK1 is predominantly expressed in the lung, bone marrow, and T cells; RSK2 is more abundant in T cells, lymph nodes, and prostate. RSK3 has high expression in the lung, brain, spinal cord, and retina. RSK4 transcript is much lower than that of other isoforms.

## 3. Regulation of RSK Isoforms

RSKs are directly or indirectly phosphorylated by several kinases in the Ras/Raf/MEK/ERK signal cascade. All human RSK isoforms have six conserved phosphorylation sites: Ser221, Thr359, Ser363, Ser380, Thr573, and Ser732, respective to RSK1 (Figure 3) [20].

In quiescent cells, RSKs are in complex with inactive ERK1/2 through kinase interaction motif (KIM) ERK-docking site. Following the stimulation of cells, ERK1/2 phosphorylate Thr573 within the CTKD activation loop leading to full activation of CTKD [12,21]. ERK1/2 might also phosphorylate Ser363 and Thr359 in the conserved linker region. The fully activated CTKD then auto-phosphorylates Ser380 within the hydrophobic motif of the linker region [11], promoting recruitment of 3′-phosphoinositide-dependent kinases-1 (PDK1) [22]. After binding, PDK1 phosphorylates Ser221 within the NTKD [23,24], leading to the full activation of NTKD and subsequent phosphorylation of the substrates [25]. NTKD also auto-phosphorylates Ser732 located next to ERK1/2 docking site, decreasing the ERK affinity for RSKs in a negative feedback loop [14]. De-phosphorylation of RSKs by protein phosphatases PP2Cdelta has been also reported as the other inactivation mechanism of RSK isoforms [26].

More studies provide evidence for alternative RSK2 activation mechanisms through the p38 MAPK and fibroblast growth factor receptor-3 (FGFR3) [27,28,29]. p38 MAPK-regulated kinases MK2/3 (MAPKAPK2/3) function as potential CTKD activation in dendritic cells [27]. FGFR3 and Scr family kinases phosphorylate RSK2 at Tyr529 to promote inactive ERK recruitment and Tyr707 to potentially disrupt the auto-inhibitory motif [28,29,30].

## 4. The Function of RSK Isoforms

Analysis of RSK1 substrates by using synthetic peptide libraries has identified the minimum consensus sequences (Arg/Lys-Xaa-Arg-Xaa-Xaa-pSer/Thr or Arg-Arg-Xaa-pSer/Thr). Although RSK1 preferentially phosphorylates this motif, there are diverse substrates containing different targeting sequences from this motif [2,31].

Through regulating extensive downstream substrates in transcriptional gene expression and directly affecting protein function, RSKs are involved in various cellular processes such as protein synthesis, proliferation, cell survival, and motility (Figure 4) [4,5,18]. In addition, RSKs have critical roles in cancer initiation by regulating cell proliferation and survival as well as cancer development/progression by regulating metastasis, cancer stem cell, and chemoresistance [6].

### 4.1. Protein Synthesis

Many findings support that RSKs are involved in translational regulation through the phosphorylation of several ribosome-associated proteins. One such way is through modulating the mammalian target of rapamycin (mTOR) pathway. RSKs phosphorylate tuberous sclerosis complex 2 (TSC2) and Raptor to promote mTOR complex 1 (mTORC1)-mediated translation [32,33,34,35]. RSKs also phosphorylate and inhibit glycogen synthase kinase-3 (GSK3), causing the stimulation of the translation initiation factor eIF2B [36]. Independent from the mTOR pathway, RSKs directly phosphorylate the eIF4B to stimulate the translation initiation of growth- and survival-related transcripts [37,38]. Recently, RSKs are also reported to promote the degradation of programmed cell death 4 (PDCD4), a negative regulator of the eIF4A helicase [39,40]. Moreover, RSKs facilitate the assembly of the translation preinitiation complex and increase cap-dependent translation by phosphorylating the 40 s ribosomal protein s6 (RPS6) [41].

### 4.2. Cell Cycle and Proliferation

There is much evidence that RSKs promote cell proliferation through modulating the components of cell cycle machinery. RSKs regulate G1 progression by phosphorylating c-Fos, GSK3, and p27^Kip1^. RSK2-dependent stabilization of c-Fos is essential to increase the cyclin D1 expression and thereby facilitate G1-S phase progression [42,43,44]. Phosphorylation on p27^Kip1^ by RSK1 suppresses its inhibition activity on the cyclin E/A-CDK2 complexes [45,46]. GSK3 normally targets c-Myc and cyclin D1 degradation. Therefore, inhibition of GSK3 activity by RSKs facilitates G1 phase progression [47,48]. RSKs also promote G2-M transition by phosphorylating and activating CDC25C involved in CDK1-cyclin B complex activation [49].

RSKs have been shown to promote cell proliferation through phosphorylating Max dimerization protein-1 (Mad1), a suppressor of Myc-mediated cell proliferation and transformation. Phosphorylated Mad1 undergoes proteasomal degradation and thereby alleviating its suppression of Myc [50]. RSKs also phosphorylate cAMP response element-binding protein (CREB), which is a critical transcription factor for cell proliferation and survival through regulation of c-Fos expression [51].

### 4.3. Cell Survival

RSK-mediated phosphorylation involves the inactivation of pro-apoptotic proteins and the activation of transcription factors controlling cell survival promotion. In response to growth factors, RSKs directly phosphorylate pro-apoptotic proteins BCL2-associated agonist of cell death (BAD) and death-associated protein kinase (DAPK) [1,52,53,54]. This results in their decreased activities and increased cell survival. In addition, RSKs inhibit caspase activity by phosphorylating the CCAAT/enhancer binding protein beta (C/EBPβ) [55]. Phosphorylated C/EBPβ creates a functional XEVD caspase inhibitory box to inhibit caspases 1 and 8 activities. In addition to the function of cell proliferation, RSK-activated CREB promotes cell survival by increasing the transcription of pro-survival genes such as B-cell lymphoma 2 (BCL2), B-cell lymphoma-extra large (BCL-XL), and myeloid cell leukemia-1 (MCL-1) [1,56].

### 4.4. Cell Motility and Metastasis

RSKs are involved in cell migration, invasion, and metastasis in various cell types through both transcriptional and protein regulation [7]. RSKs activated by ERK1/2 phosphorylate the transcriptional factor Fos-related antigen 1 (FRA1) to promote the expression of pro-metastatic genes such as Integrin, Laminin332, matrix metalloproteinase (MMP), and Rac1 [57]. RSKs also phosphorylate Filamin A and thereby promote cell migration and suppress integrin activation and cell adhesion [58,59]. Finally, RSKs phosphorylate and inactivate the SH3 domain-containing protein (SH3P2), a negative regulator of cell motility [60].

### 4.5. Cancers

The aberrant expression and activity of RSKs has been associated with uncontrolled proliferation and prolonged survival in several cancer types. In addition, RSKs are found to phosphorylate cell cycle checkpoint components such as checkpoint kinase 1 (Chk1) and Mer11 to suppress DNA damage signaling and possibly increase cancer cell chemoresistance [61,62].

There is evidence that RSK isoforms have functional differences, especially in cancer [6]. RSK1/2 promote cancer growth, proliferation, and survival. The expression or activation of RSK1/2 appears to be increased in many cancers including lung cancer [63,64], head and neck squamous cell carcinoma (HNSCC) [65], breast cancer [40,66], prostate cancer [67], leukemia [68], melanoma [39,69], multiple myeloma [28], and glioblastoma [58,70]. The expression and activation of RSKs in lung cancer inhibits cell death through inactivation of a pro-apoptotic protein BAD [64]. The loss of RSK2 function leads to decreased phosphorylation of CREB and Hsp27, and thereby substantially reduces invasiveness of HNSCC cells [65].

In contrast, RSK3/4 may act conversely as tumor suppressors. Reduced expression of RSK3/4 has been observed in various cancer types such as ovarian cancer [71], colorectal cancer [72], acute myeloid leukemia [73], and breast cancer [74,75]. Low expression of RSK4 is predictive of a poor prognosis in patients with colorectal cancer [72]. In ovarian cancer cell lines, overexpression of RSK3 was found to decease their proliferation [71]. Overexpression of RSK4 decreased the proliferation of breast cancer by accumulating cells in the G0/G1 phase [75].

Moreover, RSK4 has been shown to participate in p53-dependent cell growth arrest [76] and in oncogene-induced cellular senescence in colon and renal cell carcinomas [77]. However, it has also been reported that RSK4 is overexpressed in >50% of primary malignant lung cancers [64], and RSK3/4 can mediate tumor resistance to PI3 kinase inhibitors in breast cancer [78]. Overall, more in-depth studies are required to fully understand the role and associated molecular mechanisms of action of the different RSK isoforms in cancer.

## 5. RSK Isoforms in AML

AML is a genetically and phenotypically heterogeneous hematological malignancy characterized by the accumulation of immature myeloid blasts with peripheral blood cytopenia [79,80]. AML patients have an overall survival of less than 65% in children and 40% in adults. The current treatment options, including intensive chemotherapy and stem cell transplantation, are associated with significant morbidity and mortality [81]. Thus, it is critical to develop more effective and less toxic therapies for AML.

The Ras/Raf/MEK/ERK pathway has been reported to be constitutively activated in more than 50% of AML and acute lymphocytic leukemia (ALL) cases [82,83]. Thus, RSK isoforms play an important role in AML pathogenesis and progression. In addition, it has been observed that RSK1/2 are the predominant isoforms expressed in AML cells, whereas RSK4 has shown significantly lower expression in AML patients compared to healthy people. This suggests that downregulated RSK4 expression may lead to leukemia or negatively affect the prognosis of patients with AML [73]. We discuss below the roles of RSKs in AML pathogenesis (Figure 5) and how RSKs could be a therapeutic target for AML treatment.

### 5.1. RSKs and CREB in AML Cell Survival

Our group has previously observed that approximately 60% of AML patients express CREB at high levels, and this is associated with an increased risk of relapse and decreased event-free survival [84,85]. Similarly, the expression and phosphorylation levels of RSKs are significantly increased in pediatric AML patients associated with poor survival [86,87]. Furthermore, we have observed that RSKs phosphorylate CREB on Ser133, and then phosphorylated CREB mediates proliferation and survival of myeloid cells through induced expression of Bcl-2, cyclin A, and cyclin D [84,86]. In addition, we have recently established that RSK inhibition inhibits AML cell proliferation through the regulation of mitotic exit [87]. A potent RSK inhibitor, BI-D1870, increases metaphase arrest by preventing the metaphase/anaphase transition, followed by induced apoptosis of AML patient cells through impeded association of cell division cycle 20 (CDC20) with anaphase promoting complex/cyclosome (APC/C) and increased mitotic arrest deficient 1 (MAD2) and CDC20 binding. Moreover, BI-D1870 treatment potentiates the anti-leukemic activity of vincristine via synergistically increased mitotic arrest and apoptosis in AML cells. Therefore, our findings suggest a novel therapy that overcomes vincristine resistance to AML cells.

### 5.2. RSKs in FLT3-ITD+ Cell Survival

FLT3 is a receptor-tyrosine kinase expressed on hematopoietic progenitor cells and plays an important role in proliferation, survival, and differentiation of these cells [88,89]. The internal tandem duplication mutation in FLT3 (FLT3-ITD) is the most frequent mutation in AML found in 25–30% of cases and associated with a poor prognosis [90]. FLT3-ITD leads to constitutive activation of the Ras/Raf/MEK/ERK pathway [82,83]. As a downstream regulator of this pathway, RSKs have essential roles in the pathogenesis and myeloid lineage determination of FLT3/ITD-induced hematopoietic transformation. Activated RSK1 phosphorylates and inactivates pro-apoptotic BAD protein, preventing the apoptosis of BaF/FLT3-ITD cells [91]. Inhibition of RSK1 expression reduces BAD phosphorylation resulting in the induced apoptosis of MV4-11, a cell line harboring FLT3-ITD. Targeting RSK2 by FMK, a RSK inhibitor, attenuates cell viability and induces significant apoptosis in human primary FLT3-ITD+ leukemic cells [68]. These findings suggest that combined inhibition of FLT3 and RSKs may be a viable therapeutic strategy to cure AML patients with FLT3-ITD.

In addition, recent studies have reported that RSKs have pro-survival functions as a new target of Pim2 kinase in relapsed FLT3-ITD+ AML cells [92,93]. Pim2 is a downstream target of FLT3-ITD+ AML cells and directly contributes to FLT3 inhibitor resistance [93]. Ectopic expression of RSK2 rescues the viability of Pim2-depleted cells through the regulation of Bax expression. These support the involvement of RSK2 in AML cell survival as a downstream of Pim2 and a novel therapeutic strategy against therapy-resistant FLT3-ITD+ AML.

Furthermore, it has been reported that FLT3-ITD activates RSK1 to enhance proliferation and survival of AML cells by activating mTORC1 and eIF4 [94]. Activated RSK1/2 via FLT3-ITD and MEK/ERK pathways phosphorylate TSC2 and eIF4B in cooperation with Pim2, thus activating mTORC1/S6K/4EBP1 pathway resulting in enhanced proliferation.

### 5.3. Alternative RSK Isoform Activation in FGFR3-Activated Cells

It has been shown that RSK2 has a role in hematopoietic transformation of AML and multiple myeloma via an alternative mechanism of RSK activation [28,30]. FGFR3 directly phosphorylates Tyr529 on RSK2, which facilitates inactive ERK1/2 binding to RSK2, and consequently phosphorylates and activates RSK2 [28]. FGFR3 additionally phosphorylates Tyr707 on RSK2 that may disrupt the auto-inhibitory αL-helix motif on C-terminal [30]. Targeted inhibition of RSK2 effectively induced apoptosis in FGFR3-expressing myeloma cells, suggesting that RSK2 is a critical signaling effector in FGFR3-mediated hematopoietic transformation.

### 5.4. RSKs in the Resistance to As_2_O_3_ Treatment

Arsenic trioxide (As_2_O_3)_ treatment is an effective therapy for acute promyelocytic leukemia (APL) which is a subtype of AML, but shows no significant clinical activity in other non-APL subtype refractory or relapsed AML cases [95]. Recent studies have implicated that RSK1 is involved in the resistance of AML to As_2_O_3_ [96,97]. RSK1 is phosphorylated and activated during As_2_O_3_ treatment in different AML cell lines. This suggests that RSK1 counteracts As_2_O_3_-dependent anti-leukemic response by being activated in a negative feedback regulatory manner. Combined treatment of RSK1 inhibitor with As_2_O_3_ was found to result in more potent suppression of leukemic cells. These results suggest that RSK1 is a potentially important target to enhance the anti-leukemic properties of As_2_O_3_.

### 5.5. RSKs with Inhibition of SHH Signaling

Sonic Hedgehog (SHH) signaling is implicated in drug resistance for a range of human cancers [98]. A recent study showed that RSK inhibition overcomes resistance to inhibition of the SHH pathway in pediatric medulloblastoma [99]. In addition, SHH signaling has been found to play a role in the self-renewal of leukemia stem cells (LSCs) for CML and multiple myeloma [100,101,102]. These findings suggest that the combination of RSK and SHH inhibitors may be a complementary anti-leukemia strategy especially through regulating LSCs.

## 6. Current RSK Inhibitors

As RSKs play important roles in various cancers including AML through disease-related signaling pathways, RSK inhibition provides a promising therapeutic strategy for many diseases. So far, various RSK inhibitors are reported to target the NTKD or CTKD on RSK isoforms (Table 1) [103]. However, none of them is currently useful for in vivo use in cancer models. Furthermore, current RSK inhibitors target more than one RSK isoforms, suggesting the limitation of their efficacy as anticancer agents. As isoform-specific function of RSK is suggested in various cancers, more efficient and specific RSK inhibitors need to be developed for molecular therapy medicine in the future.

### 6.1. SL0101

The flavonol rhamnoside SL0101 is a cell-permeable kaemperfol glycoside isolated from the tropical plant *Fosteronia refracta*. SL0101 is the first identified pan-RSK inhibitor. This is shown to target the NTKD of RSK1 and RSK2 in the nanomolar range (IC_50_ of 89 nM at 100μM ATP for RSK2) while having no significant effects on other AGC kinases [57,104]. SL0101 treatment has been shown to impair the growth of MCF7 breast cancer cell but not of MCF-10A normal breast epithelial cells [104]. However, this has shown a higher EC50 in vivo and lower stability. Although several analogues have been developed [105], poor pharmacokinetic (PK) properties and off-target effects have limited their development.

### 6.2. BI-D1870

The dihydropteridinone BI-D1870 is the pan-RSK inhibitor as an ATP antagonist to target NTKD [106]. BI-D1870 completely abrogates activities of all RSK isoforms and has selectivity for RSKs relative to other AGC kinases [107]. However, BI-D1870 also significantly inhibits polo like kinase 1 (PLK1) with similar potency and Aurora B, maternal embryonic leucine zipper kinase (MELK), PIM3, mammalian sterile 20-like kinase 2 (MST2), and GSK3β activities at 10- to 100-fold higher concentration [107].

We have observed that 5μM of BI-D1870 specifically inhibits the proliferation of AML primary cells as well as HL60 AML cell line [87]. Among four RSK isoforms, BI-D1870 shows 2-fold lower potency toward RSK3 and RSK4 as compared to RSK1 and RSK2 (IC_50_ of 15 nM for RSK4, 18 nM for RSK3, 24 nM for RSK2, and 31 nM of RSK1) [106]. This suggests the structural divergence of RSK isoforms that could be exploited for design of isoform-specific inhibitors. Similar with SL0101, BI-D1870 has shortcomings including poor PK profiles and non-specific interaction. There are a few structure–activity relationship (SRA) studies based on BI-D1870 to improve the limitations: substituted pteridinones such as the difluorophenyl pyridine derivatives (LJH685, LJI308) and a series of substituted pteridinones and pyrimidines [108].

### 6.3. LJH685 and LJI308

LJH685 and LJI308 are difluorophenyl pyridine derivatives of BI-D1870. These have IC_50_ of 4–13 nM range against all RSK isoforms and, more importantly, have much fewer off-target effects than BI-D1870 [109]. However, it has been reported that LJH685 exhibits high clearance, short plasma half-life, and mild tissue distribution in rats, which should be addressed in further studies [110].

### 6.4. BIX02565

BIX02565 has been reported to target RSK2 with IC_50_ of 1.1 nM [111]. It is unclear whether BIX02565 inhibits other RSK isoforms. However, this also has poor selectivity as it also inhibits leucine rich repeat kinase 2 (LRRK2), rearranged during transfection (RET), ded2-like kinase 2 (CLK2), FLT3, and platelet-derived growth factor receptor (PDGFR) [112]. Furthermore, BIX02565 has shown to cause severe decrease in mean heart rate and arterial pressure in a mouse model [113].

### 6.5. BRD7389

BRD7389 is a RSK inhibitor that induces insulin expression and differentiation of pancreatic alpha into beta cells [114,115]. However, this inhibits RSKs with relatively higher IC_50_ of 1.5 μM for RSK1, 2.4 μM for RSK2, and 1.2 μM for RSK3 [114]. Moreover, BRD7389 also inhibits CDK5, death-associated protein kinase-related apaptosis-inducing protein kinase 1 (DRAK1), FLT2, PIM1, and protein kinase G 1 α (PKG1α) in very lower IC_50_ of 2.8–6.5 μM.

### 6.6. FMK

Pyrrolopyrimidine FMK (fluoromethyl ketone) is an irreversible RSK inhibitor that binds to cysteine residue on the ATP-binding pocket of CTKD [116]. This inhibits RSK phosphorylation on Ser386 and downstream singling with IC_50_ of 15 nM for RSK2 [117]. However, FMK also inhibits Lck, Src, EphA2, and S6K1 with greater concentration [117]. Besides, FMK inhibits only the activating process of RSK, as CTKD is not necessary once the NTKD is active through other kinases such as p38.

### 6.7. PMD-026

PMD-026 is the first orally bioavailable small molecule inhibitor targeting RSK. It has been developed to treat triple negative breast cancer (TNBC) and is currently in Phase1/1b clinical trial for advanced breast cancer [118]. PMD-026 demonstrates high specificity for the four RSK isoforms in vitro (IC_50_ 0.7–2 nM) with good selectivity as well as in vivo efficacy in mouse xenograft tumor models of TNBC. Moreover, PMD-026 did not cause any apparent cardiotoxicity, neutropenia, or ocular toxicity in mice and dogs, setting this inhibitor apart from other RSK inhibitors. In addition, preclinical data show that PMD-026 is effective alone or in combination with conventional chemotherapies. PMD-026 in combination with chemotherapy has the potential to become a platform technology for a wide range of refractory cancers.

## 7. Conclusions

The roles of RSKs in diverse cellular processes are context-dependent and complex. RSKs have important roles in AML pathogenesis, so they are attractive therapeutic targets for AML. However, current RSK inhibitors are not useful as cancer therapies due to limitations such as off-target effects and poor PK properties. Emerging evidence suggests that RSKs have isoform-specific functions. Thus, a critical goal for future research is to develop not only more RSK inhibitors, but isoform-selective inhibitors. Further studies may aid in the identification of RSK substrates and evaluate the role of RSK isoforms and RSK inhibition as anti-leukemic strategy.

## Figures and Tables

**Figure 1 biomedicines-09-00726-f001:**
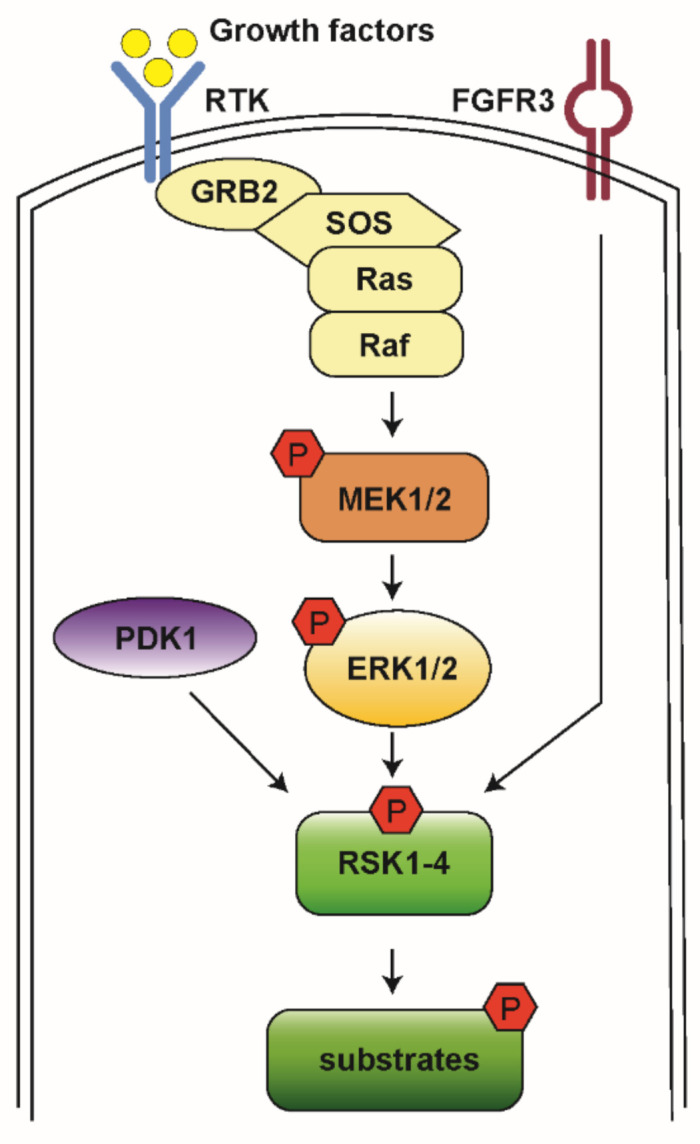
A schematic model for RSK activation. When RTK is stimulated by growth factor, it activates the docking proteins GRB2 and SOS. SOS triggers Ras to exchange guanosine diphosphate (GDP) to guanosine triphosphate (GTP) and then to become activated. Ras activates Raf kinases, which phosphorylate MEK1/2, ERK1/2, and RSK1-4. Then, RSKs phosphorylate various downstream substrates to mediate diverse cellular processes.

**Figure 2 biomedicines-09-00726-f002:**
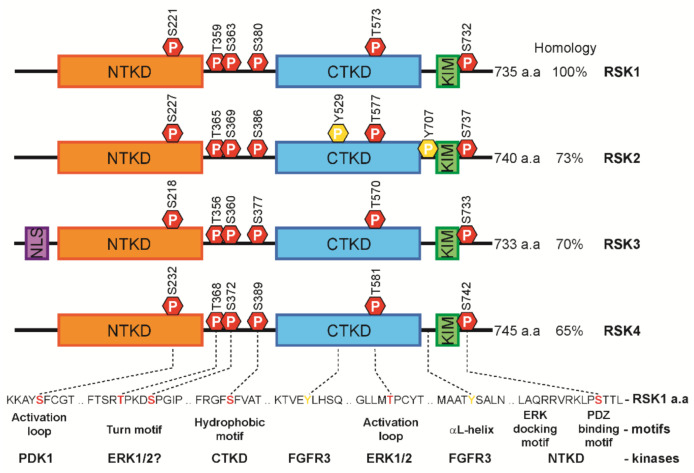
The domain structure of four RSK isoforms in human. RSKs exhibit two functional domains NTKD and CTKD, which are connected by a linker region. RSKs have six conserved phosphorylation sites. C-terminal tail contains an ERK1/2-docking domain called KIM motif and PDZ-binding motif.

**Figure 3 biomedicines-09-00726-f003:**
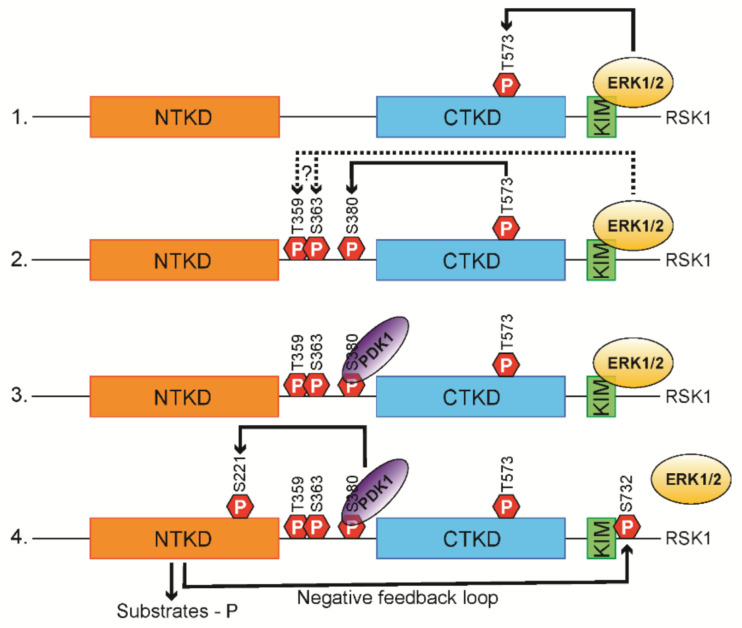
A schematic model of RSK1 activating process. Extracellular signals activate ERK1/2, which phosphorylates Thr573 in CTKD. The activated CTKD autophosphorylates Ser380 at the hydrophobic motif. In addition, ERK1/2 might carry out the phosphorylation for Thr359 and Ser363. A constitutively active Ser/Thr kinase PDK1 binds at the phosphorylated Ser380 and phosphorylates Ser211 in the NTKD, which leads to the full activation of RSKs and subsequent phosphorylation of the various substrates. This is followed by the phosphorylation of Ser749 by the NTKD, decreasing the ERK1/2 affinity for RSKs in a negative feedback loop.

**Figure 4 biomedicines-09-00726-f004:**
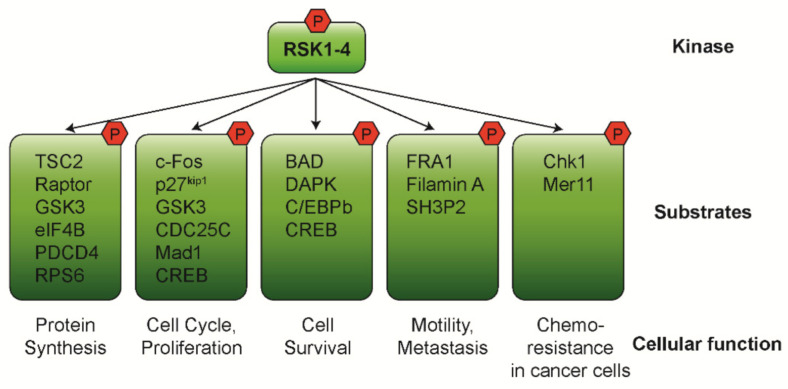
Molecular targets and cellular function of RSK isoforms. RSKs control diverse cellular processes by regulating extensive downstream targets.

**Figure 5 biomedicines-09-00726-f005:**
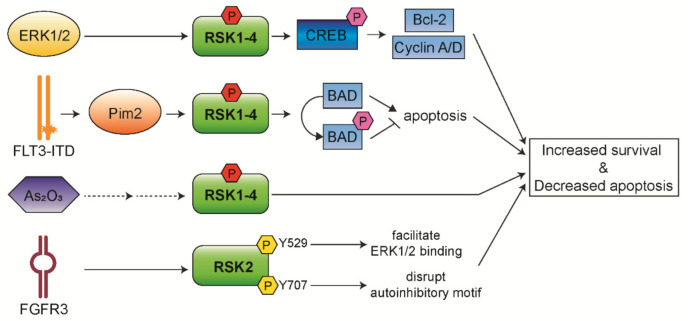
Functional mechanisms of RSK isoforms in AML. RSKs control survival and apoptosis of leukemic cells through aberrantly activated upstream signaling.

**Table 1 biomedicines-09-00726-t001:** Summary of pharmacological inhibitors of RSK isoforms.

Inhibitors	Structure	Target	IC_50_	EC_50_
SL0101	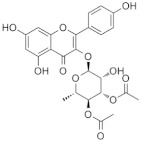	CKTD	89 nM in RSK2	50 μM
BI-D1870	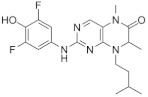	CKTD	31 nM in RSK124 nM in RSK218 nM in RSK315 nM in RSK4	1 μM
LJH685	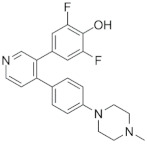	CKTD	6 nM in RSK15 nM in RSK24 nM in RSK3	730–790 nM
LJI308	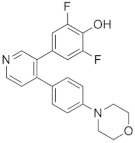	CKTD	6 nM in RSK14 nM in RSK213 nM in RSK3	200–300 nM
BIX02565	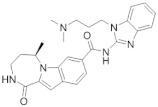	CKTD	1 nM in RSK2	N.A.
BRD7389	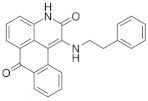	CKTD	1.5 μM in RSK12.4 μM in RSK21.2 μM in RSK3	N.A.
FMK	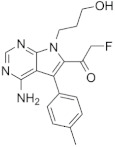	NKTD	15 nM in RSK2	200 nM
PM-026			0.7–2 nM in RSK2

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
