# Peer review of "RSK Isoforms in Acute Myeloid Leukemia"

_biomedicines, 2021, doi:10.3390/biomedicines9070726_

Round 1

Reviewer 1 Report

This review is well-written, comprehensive, and very well organized.   It will certainly make significant contribution for our understanding of the function of RSKs in AML and other cancers.  The figures were well-made and enjoyable.

The only concern is that the authors miss to discuss the first RSK Inhibitor undergoing Phase 1/1b Clinical Trial, PMD-026, for advanced Breast Cancer.  This is a very important drug and should be added to section 6: “Current RSK inhibitors”.

Author Response

1. The only concern is that the authors miss to discuss the first RSK Inhibitor undergoing Phase 1/1b Clinical Trial, PMD-026, for advanced Breast Cancer.  This is a very important drug and should be added to section 6: “Current RSK inhibitors”.

-> Response 1: Thank you for this great suggestion. We have now added discussion about PMD-026 in section 6: "Current RSK inhibitors"

Reviewer 2 Report

Although the figures are representative for what the authors want to convey, it can be seen that they have been made in powerpoint. Please reach out and make them in a more professional manner.

Please expand the table regarding the RSK inhibitors so that it is easier to see them.

Otherwise I consider that the current review would be appropriate to be published after these minor changes.

Author Response

1. Although the figures are representative for what the authors want to convey, it can be seen that they have been made in powerpoint. Please reach out and make them in a more professional manner.

-> Response 1: Thank you for this suggestion. We have now made the figures using Illustrator program and replaced the old figure.

2. Please expand the table regarding the RSK inhibitors so that it is easier to see them.

-> Response 2: Thank you for this suggestion. We have now expanded the table and replaced the old table.